# Evaluation of Multimodal and Multi-Staged Alerting Strategies for Forward Collision Warning Systems

**DOI:** 10.3390/s22031189

**Published:** 2022-02-04

**Authors:** Jun Ma, Jiateng Li, Hongwei Huang

**Affiliations:** School of Automotive Studies, Tongji University, No.4800, Cao-an Road, Shanghai 201804, China; majun.tongji@foxmail.com (J.M.); 2111304@tongji.edu.cn (H.H.)

**Keywords:** intelligent connected vehicle, active safety, multimodal warning, multi-staged warning, driving simulator, eye-tracking analysis

## Abstract

V2X is used for communication between the surrounding pedestrians, vehicles, and roadside units. In the Forward Collision Warning (FCW) of Phase One scenarios in V2X, multimodal modalities and multiple warning stages are the two main warning strategies of FCW. In this study, three warning modalities were introduced, namely auditory warning, visual warning, and haptic warning. Moreover, a multimodal warning and a novel multi-staged HUD warning were established. Then, the above warning strategies were evaluated in objective utility, driving performance, visual workload, and subjective evaluation. As for the driving simulator of the experiment, SCANeR was adopted to develop the driving scenario and an open-cab simulator was built based on Fanatec hardware. Kinematic parameters, location-related data and eye-tracking data were then collected. The results of the Analysis of Variance (ANOVA) indicate that the multimodal warning is significantly better than that of every single modality in utility and longitudinal car-following performance, and there is no significant difference in visual workload between multimodal warning and the baseline. The utility and longitudinal driving performance of multi-staged warning are also better than those of single-stage warning. Finally, the results provide a reference for the warning strategy design of the FCW in Intelligent Connected Vehicles.

## 1. Introduction

Traffic accidents cause huge casualties and economic losses and have become a serious problem for all countries. The Ministry of Public Security of China announced that motor vehicle-related accidents accounted for 86.8% of traffic accidents in 2019. According to the National Highway Traffic Safety Administration, rear-end accidents account for about 30% of all traffic accidents [1]. Jamson et al. [2] showed that most accidents could be avoided if drivers were alerted and enabled to take avoidance measures one second before a rear-end collision occurred. A forward collision warning system (FCW) has been shown to be effective in alerting drivers in emergencies and then helping them to react more quickly, ultimately helping them to avoid collisions [3].

In order to avoid forward collisions, an FCW system should provide timely and accurate alerts to the driver, and the alerts should not significantly interfere with driving performance. As a result, FCW is designed to meet these requirements through two alerting strategies: sensory channels and alert levels.

The multimodal warning is to convey the information of safety, efficiency, and service of the intelligent connected vehicle to the driver in a timely and clear manner through different sensory channels of the driver [4]. The advantage is more evident in the fact that multimodal warning can also be delivered to the driver through tactile and auditory redundant channels when the driver’s eyes deviate from the interface of warning [5]. For the comparative study of different warning modalities, the existing studies are mainly concerned with the utility of the warnings and the effect on the subjective load and driving performance of the driver. One study investigating the utility of tactile warning for rear-end accident prevention found it to be superior to both auditory and visual warning by the metric of reaction time in a simulated driving environment [6,7]. In addition to the comparison of single-modality warnings, it has been noted that the driver’s reaction time to an impending collision event is reduced when an auditory alert based on speech or tone is provided before a tactile channel alert [8]. Francesco et al. [9] compared the utility of auditory warning, vibration warning, and multimodal warning, and the multimodal warning resulted in the shortest brake reaction time in an impending collision situation. Haasd and Erp pointed out that the two dual-modality warnings, visual plus auditory and visual plus tactile, were more effective than visual warning alone through a comparative analysis of utility metrics [10]. However, it has also been pointed out that multimodal warning can lead to an increase in driver subjective load, which in turn causes a decrease in the utility of the warning [9]. Therefore, when evaluating sensory channel reminder strategies, their effects on driver subjective load need to be considered. In addition to this, Li Lin used driving performance indicators (vehicle speed, acceleration, lateral displacement, etc.) as objective evaluation parameters [11]. However, the current evaluation of warning modalities lacks the measurement of the driver’s visual workload. This is because visual and auditory warnings may cause inattentive driving and eventually lead to visual distraction. Therefore, for the evaluation of different modalities, utility is the main objective indicator, but also considers its impact on driving performance and visual load, and finally adds subjective evaluation. On the other hand, the current research on the evaluation of visual warnings for FCW mainly takes the dashboard as the output equipment. With the popularization of Head-Up Display (HUD) in passenger cars, they are adopted as an emerging device to not only provide drivers with driving aids, but also reduce the visual workload caused by warning messages [12]. However, there are few evaluation tests on FCW using HUD as a visual output. In this paper, we add HUD as a visual warning modality for evaluation.

The multi-staged warning provides drivers with continuous and graded warnings based on the division of the pre-crash warning scenario so that drivers can take appropriate measures according to the urgency of the alert [13]. For the evaluation of warning stages, a prior study designed a timing strategy for multi-staged warning based on the relative speed, and also evaluated the utility of this system using subjective indicators [14]. Since the multi-staged FCW system is still in the stage of preliminary application, there are fewer studies on the evaluation of this warning strategy. In order to have a more comprehensive understanding of this strategy, more objective indicators are needed for evaluation in addition to the subjective utility.

In this paper, in order to compare and evaluate the differences in the warning strategies of FCW systems, we first designed the warning modality as a visual warning, auditory warning, tactile warning, and multimodal warning and then selected the HUD as the visual modality for designing a two-stage FCW. Subsequently, an open-cab cockpit was used as a driving simulator to collect driving performance data and eye-tracking data during the simulated driving. Finally, the data were processed by ANOVA to compare and evaluate two types of warning strategies for FCW, namely, warning modalities and warning stages, in four dimensions: utility, driving performance, visual workload, and subjective evaluation.

## 2. Materials and Methods

In the present experiment, the setting of FCW scenario is that the host vehicle (driven by the participants) is in the same lane as the target vehicle (controlled by the computer) and there is a collision hazard between them. The FCW system then sends an alert through the interactive channel to remind the driver of the existence of a collision hazard, and finally the driver takes measures to avoid the collision [14]. The FCW scenario belongs to the high-priority and high-urgency scenario, so the FCW should ensure a strong warning effect and clear warning content. The warning strategy in this scenario mainly includes different warning modalities and warning stages. The modalities are mainly divided into a single and multimodal channel of warning, and multi-staged warning includes two levels (collision pre-warning and collision warning).

### 2.1. The Design of Multimodal and Multi-Staged Forward Collision Warning

The visual channel is a suitable way to deliver complex information to the driver, which is mainly continuous information (in-car navigation), low-priority information (in-car media), and warning information (vehicle status information) [15]. If the information is presented on the center console or dashboard, the human–machine interaction (HMI) will take up the driver’s visual resources, taking his/her eyes off the road and causing visual distraction. According to a naturalistic driving study, engaging visual–manual activities with in-car HMI is linked to a substantial 5-fold increase in collision risk compared to driving without visual distraction [16]. The types of equipment that present visual information in the cockpit of an intelligent connected vehicle mainly include the HUD, the instrument panel, and the center console. In the forward collision warning scenario, the design of the HUD is integrated into the SCANeR studio^®^ scenario, and the interaction design of the dashboard is completed and presented on the two tablets through Protopie^®^.

The auditory channel quickly attracts the driver’s attention and delivers the information with short, high-priority content. The auditory channel provides timely feedback in the form of a tone or voice to the driver, who then needs to quickly take measures for driving safety. In addition to timeliness, the use of stereo sound warnings to provide directional information to the driver can effectively reduce the time to detect collision-hazard vehicles [17]. Since forward collision warning is a safety-related alert and requires the driver to maneuver the vehicle in time, the alert tone was chosen as the auditory warning modality. The alert tone on the two tablets was designed by Protopie^®^. The sound pressure level of the alert tone is 75 dB and the frequency is 2000 Hz, which complies with ISO [18] and SAE [19] standards.

The haptic channel has similar utility to the auditory channel warnings [4], and both are used in combination to provide perceptible warnings to the driver in high-noise environments. The haptic warning alerts the driver through vibration signals generated by the seat, seat belt, steering wheel, and foot pedals. For the timeliness of vibration alerts, Petermeijer et al. [20] suggest that haptic and visual alerts can guide a distracted driver to take over during the autonomous driving takeover phase. Due to the force feedback steering wheel of the simulated cockpit, seat vibration was adopted to remind the driver. The design of the multimodal FCW cockpit of the intelligent connected vehicle is shown in Figure 1.

The differentiated design of the warning stages is mainly reflected as a gradual warning process. This experiment sets two warning stages: collision pre-warning and collision warning. Lin Li et al. [11] pointed out that the traditional single-level emergency warning strategy is not the optimal design solution for advance collision warning. Meanwhile, the results of Winkler et al. pointed out that an efficient crash warning design should not only warn the driver when a collision is imminent, but also provide a pre-warning to the driver when the possibility of a collision exists [14].

### 2.2. The Trigger Timing Design of Forward Collision Warning

In addition, the timing of the warning is also of great importance because it is necessary to provide the driver with enough time to detect the dangerous situation and then take measures. The Time-To-Collision time (TTC) was selected as the trigger parameter of the forward collision warning. Three indicators (minimum speed of the host vehicle, the minimum relative speed between the two vehicles, and the host vehicle deceleration threshold of two vehicles sharing the same speed without colliding) were used as the restriction of the warning [21]. There are two warning stages in the experiment: collision pre-warning stage and collision warning stage, so the warning timing design is performed for these two stages. This system starts the collision warning when the TTC is less than 3 s, and the collision pre-warning is triggered when the TTC is greater than or equal to 3 s and less than 5 s. For the restrictions of the FCW: the minimum speed is less than 4.17 m/s, the relative speed of the two vehicles is less than 1 m/s, and there is no collision deceleration below its threshold value of 6 m/s^2^ when the two cars share the same speed. If one of the above three conditions is met, neither collision pre-warning nor collision warning is provided. The formula for calculating TTC is shown in Equation (1), while the calculation of the no-collision deceleration when the two cars share the same speed is shown in Equation (2).
(1)$$\mathrm{TTC}=\frac{d_{\mathrm{CT}}\left(t\right)}{v_{C}\left(t\right)-v_{T}\left(t\right)}$$
(2)$$a_{S}=\frac{{\left[v_{C}\left(t\right)-v_{T}\left(t\right)\right]}^{2}}{2\times \left[d_{\mathrm{CT}}\left(t\right)-d_{R}\left(t\right)\right]}+a_{T}\left(t\right)$$

As shown in the equations, d_CT_(t) is the real-time distance between the front-most end of the vehicle and the last end of the target vehicle in the same lane; v_C_(t) is the real-time speed of the vehicle; v_T_(t) is the real-time speed of the target vehicle; d_R_(t) is the reduction in the distance between the two vehicles during the driver’s reaction time to the warning signal, and the driver’s reaction time is taken as 0.6s [21]; a_T_(t) is the real-time acceleration of the target vehicle.

### 2.3. The Simulated Driving Scenario Design of Forward Collision Warning

Simulated driving is an effective experimental method to obtain driver behavior data. Its advantages are mainly reflected in the safety and repeatability of the experimental process [22]. In this experiment, the forward collision warning scenario is built by SCANeR studio^®^, and the total length of the route is 2856.85 m. The driving route includes urban, suburban, and high-speed roads, where the driving scenarios mainly include right turn at the intersection, left turn at the intersection, and straight ahead at the intersection. The trigger of forward collision warning is developed by script editing of SCANeR studio^®^. The open-cab cockpit is modified based on a Fanatec^®^ simulated cockpit (Fanatec, Landshut/Bavaria, Germany: ClubSport Wheel Base V2.5, ClubSport Pedal V3 Inverted, and RennSport Cockpit). Three 43-inches screens were combined together to display the simulated driving scenarios, and they provided the drivers a 94° field of view. Moreover, a tablet was installed in place of the instrument panel in accordance with the position of production cars. The seat vibration is implemented by controlling the switch of the vibration motor mounted on the seat via Transmission Control Protocol (TCP). The relevant signals from the steering wheel, brake pedal, and acceleration pedal are transmitted to the ACQUISITION module of the SCANeR to control the vehicle in the simulated scenario. In addition, the front vehicle in the scenario was controlled by the computer in TRAFFIC module of SCANeR.

### 2.4. Data Acquisition and Procedure

The sources of experimental data contain simulation data, eye-tracking data, and subjective data. There are two main types of simulation data: vehicle-related data, and scenario-related data. Vehicle-related data are the kinematic and status parameters of the vehicle, including the speed, acceleration, status of the foot pedals, etc. The scenario-related data are the relative location of the front and host vehicle in the simulated driving system, including relative coordinate of the two vehicles, lane deviation of the host vehicle, etc. The eye-tracking data (fixations position and fixation duration) of the participants are collected by the eye-tractor developed by Pupil Labs^®^. The simulation data were sampled at 20 Hz and eye movement data were sampled at 50 Hz. Subjective evaluations of different warning strategies were collected in the form of subjective questionnaires.

A total of 30 drivers participated in the experiment (15 males and 15 females), and all drivers were between 21 and 31 years old (mean age 25.5, standard deviation 1.9). Each participant had a minimum of 2 years of driving experience and a weekly driving frequency of at least three times. All of the participants were able to adjust to dynamic virtual scenarios and had a vision of at least 4.8 without the need for glasses. Our trials do not risk the physical or emotional safety of the participants because we employed a driving simulator and the eye-tracking gear is akin to glasses. Furthermore, each participant is required to sign an informed consent form prior to the experiment. Each participant received a daily compensation of $500.

Prior to the start of the experiment, drivers underwent a pretest to become familiar with the simulated driving. If the driver felt uncomfortable during the test, we immediately stopped the experiment. During the test, participants were instructed to follow the car in accordance with their daily driving habits. If the relative distance was too far (TTC > 8 s), the experimenter would remind the driver to keep a sufficient following distance. When the front car passed the trigger point (braking trigger point), it would immediately decelerate to 20 km/h at a deceleration of 7 m/s^2^, and then participants needed to brake appropriately quickly to avoid a collision. Then, the front car kept a speed of 20 km/h till the next trigger point (normal driving trigger point) and then it was driven normally again. Each driver was required to drive the entire route 7 times; the first trip was a baseline test, followed by 6 tests (five warning modalities and one multi-staged warning). Except for the baseline test, the order of the other experiments was randomly arranged. The procedure of the driving experiment is shown in Figure 2. Data were collected at the beginning of the simulated test, and the end of data collection was when the experiment ended. At the end of each test, every participant was required to complete a subjective evaluation questionnaire.

### 2.5. Variable Interpretation and Analysis of Variance

#### 2.5.1. Independent Measures

Warning modality and warning stage are two independent measures. The warning modality mainly includes three sensory solutions: visual warning, auditory warning, and tactile warning. The visual warning is presented via two visual interaction methods: head-up display and dashboard. The auditory warning is implemented by the warning tones. The tactile warning is accomplished through seat vibrations. The multimodal warning is a combination of all the above modalities. At the stage of collision warning, a comparison experiment between different types of modality and multimodal warning modality is conducted to control the effect of the warning stages on the results. For the warning stages, two warning strategies were recognized as independent measures: collision pre-warning (TTC = 5 s) and collision warning (TTC = 3 s). The HUD was selected as the warning modality to control the effect of the modalities so as to compare and evaluate the differences in warning stages.

#### 2.5.2. Dependent Measures

The dependent measures involve four types of metrics: utility, driving performance, visual load, and subjective evaluation. The utility is quantified by the driver reaction time. The moment when the TTC reaches its threshold (single-stage: TTC = 3 s; multi-staged: TTC = 5 s) is the starting point for the reaction time. When the driver brakes, that moment was regarded as the end of the reaction time. Driving performance includes longitudinal car-following performance (time-to-collision) and lateral driving performance (standard deviation of lane departure). For the calculation of TTC, its threshold is 3 s when the independent variable is the warning modality. When we compare different warning stages, the TTC threshold is 5 s. The dependent variable (TCC) is the average of all TTCs between two adjacent thresholds. The standard deviation of lane departure (SDLD) is the standard deviation of the distance from the center of the vehicle to the center of the lane between the two thresholds. The measurement of visual workload is the off-road fixation duration (OFD). It is the average of all the fixation durations within two adjacent TTC thresholds, where the driver’s sight deviates from the road ahead and forms fixations. The OFD metric is recommended by international standards and has also been used in prior studies to measure visual distraction [23,24,25]. The subjective dependent measures include ease of perception, utility, information clarity, trust, and user experience.

#### 2.5.3. Analysis of Variance

Analysis of variance (ANOVA) was selected to assess the differences in the utility, driving performance, and visual load between the warning modalities and warning stages of the forward collision warning strategy. ANOVA was implemented by RStudio (R Core Team 2020) with the significance level α (0.05). The data were first subjected to a pre-test to verify the suitability for ANOVA, and the results of the pre-test are shown in Table 1. Firstly, the normality of the residuals between groups of the data was tested by the Shapiro–Wilk Test. For the results of the test, when the W values were close to 1 and all *p* values were greater than 0.05, the original hypothesis was rejected, and the data obeyed normal distribution. A Levene’s Test was performed to test homogeneity of variance of the residuals between the groups of data. If all *p* values in the results are greater than 0.05, the original hypothesis is rejected and the data are consistent with homogeneity of variance of the residuals between the groups of data.

## 3. Results and Discussions

ANOVAs were conducted on the reaction time, time-to-collision, the standard deviation of lane departure, and off-road fixation duration of 30 participants. The warning modality and warning stage were respectively used as the main effects of the one-way ANOVA. Partial eta-squared (Ƞ^2^) was applied to measure the sample effect size. The effect sizes will be interpreted as small for Ƞ^2^ ≥ 0.04, medium for Ƞ^2^ ≥ 0.25 and large for Ƞ^2^ ≥ 0.64. To further compare the group differences between different warning modalities and stages, the Tukey HSD test was selected to complete a post hoc test on the results of the ANOVA. The warning modality and warning stage were unordered categorical variables, with a total of six categories (including the baseline test: no forward collision warning) and two categories of warning stages.

### 3.1. Utility

The results of the ANOVA for driver reaction time showed that the difference in reaction time between modalities was statistically significant (F (5, 174) = 26.92, *p* < 0.001, Ƞ^2^ = 0.44). As shown in Table 2, the multimodal warning had the shortest reaction time and its utility was the best. As a result, for the forward collision warning, there is variability in the utility of different warning modalities and the optimal solution can be adopted in the design of warning modality. The results of the post hoc test in Figure 3 showed that the reaction time of all warning strategies is significantly shorter than the baseline (no warning), and this result is also consistent with the prior study [3]. In addition, there was no significant difference between the three warning modalities (multimodal warning, HUD, and seat vibration). Among the single warning modality, HUD was better than the seat vibration, followed by the auditory warning. However, the differences among the three single warning modalities were not significant. The dashboard as a visual warning modality has the longest reaction time leading to its poor utility. A prior study [26] also suggested that the dashboard is not a suitable bridge for displaying information related to driving safety. This result may be explained by the fact that the dashboard displays too much vehicle status information for the driver to discover the forward collision warning in time, thus increasing the reaction time. In contrast to the findings of Katharina et al. [27] and J.J. Scott [6], we introduced HUD as a visual modality to this study. One unanticipated finding was that it has a shorter reaction time than the tactile and auditory warnings. Compared to the tactile and auditory alerts, the HUD makes it possible for the driver to directly identify the vehicles about to crash in front of the driver’s field of view. As a result, it allows the driver to quickly detect and understand the information of FCW and ultimately brake to avoid the impending danger. Compared to the multimodal warning, the HUD is slightly worse in utility. This may be due to the fact that when drivers are visually distracted, they do not receive HUD warnings in time, which leads to longer reaction times. Therefore, when one of the human sensory channels (visual, auditory, or tactile channel) is occupied, the multimodal warning can still convey the warning information to the driver via other available modalities. For the comparison of the different strategies of warning stages, the reaction time of the multi-staged warning was significantly lower than that of the single-stage warning (F (1, 58) = 215.8, *p* < 0.001, Ƞ^2^ = 0.79).

### 3.2. Driving Performance

The results of ANOVA indicated that there was a significant difference in the mean time-to-collision for the different warning modalities (F (5174) =111.6, *p* < 0.001, Ƞ^2^ = 0.76), and there was no significant difference in the standard deviation of lane departure (F (5174) = 0.16, *p* = 0.977). This result indicates that the difference in warning modality significantly affects the driver’s longitudinal car-following performance. However, there is no significant effect on driving performance for lateral control of the vehicle. Compared with the other content of distraction study (e.g., IVIS interaction, in-car phone use) [28], this result can be partly explained by the fact that the duration of the warning interaction was not long enough to affect the lateral control of the vehicle. As shown in Table 3, the longitudinal car-following performance of the multimodal warnings was significantly better than the other single-stage warning. In Figure 4, the post hoc test results indicate that these differences are significant. As shown in Table 3, taking the heads-up display as the warning modality, the longitudinal car-following performance of the multi-staged warning was significantly better than that of the single-stage warning, and the difference was statistically significant (F (1, 58) = 137.3, *p* < 0.001, Ƞ^2^ = 0.70). The multi-staged warning strategy provides the warning information in advance for the driver to control the car-following distance and match it with the current speed [29]. As for the lateral vehicle control, the difference in warning stages was not significant (F (1, 58) = 2.478, *p* = 0.121)

### 3.3. Visual Workload

The results of ANOVA indicated that the difference in off-road fixation duration between warning modalities was significant (F (5, 174) = 45.5, *p* < 0.001, Ƞ^2^ = 0.61). As shown in Table 4, the visual workload from the dashboard warning was significantly higher than that of other modalities. The Pairwise comparison of off-road fixation duration for different warning modalities by post hoc test in Figure 5 indicated the statistical significance of the difference between a dashboard and other warning modalities. In addition, the difference in visual workload of the two strategies in warning stages was not significant (F (1, 58) = 0.028, *p* = 0.868).

### 3.4. Subjective Evaluation

The results of the user-subjective ratings are shown in Figure 6. The multimodal warning had the highest score in subjective utility, as they use a combination of multiple modalities to ensure that the distracted driver clearly receives the alert. However, the score of user experience for the multimodal warning was low. A possible explanation for this might be that the redundant modalities of warning can cause tension and discomfort to the driver. For the difference in warning stages, the subjective scores on the dimension of utility are opposite to the objective data results. Although multi-staged reminders can provide advance warnings, they increase the frequency of warnings and the chance of false or nuisance warnings. As a result, it leads to user annoyance, decreased trust, and the perception that the warning is ineffective [30].

## 4. Conclusions

This paper evaluates forward collision warning strategies through a driving simulator to investigate the differences between warning modalities and warning stages under four metrics: utility, driving performance, visual workload, and subjective evaluation. In the comparison of warning modalities, the multimodal warning resulted in the shortest driver reaction time. This warning strategy improves drivers’ longitudinal car-following performance. Moreover, in the comparison between the multimodal warning and the baseline (no warning is given), there is no significant difference in vehicle lateral control and visual workload for multi-channel reminders. Thus, it indicates that the multimodal warning has no significant negative effect on lateral control of the vehicle or increase in visual workload. Moreover, the overall results of the subjective evaluation can reflect the acceptance of the multimodal warning by users. For warning stages, multi-staged warning outperformed single-stage warning in terms of objective utility and longitudinal car-following performance, yet the user experience, subjective utility, and trust in multi-staged warning was worse. In this study, an open-cab cockpit of an intelligent connected vehicle was designed. Moreover, head-up display and seat vibration were introduced in the study as emerging warning modalities, and multi-staged warnings were added as warning strategies. The results of the study provide guidance for the design and evaluation of FCW warning strategies. Due to the simulator-based study, it is recommended that a real-world driving test of the FCW should be accomplished in the future to compare and validate results of the present study. Future study could also integrate various driving scenarios (e.g., distracted driving scenario) in experiment design to explore whether it can be further effective in different scenarios.

## Figures and Tables

**Figure 1 sensors-22-01189-f001:**
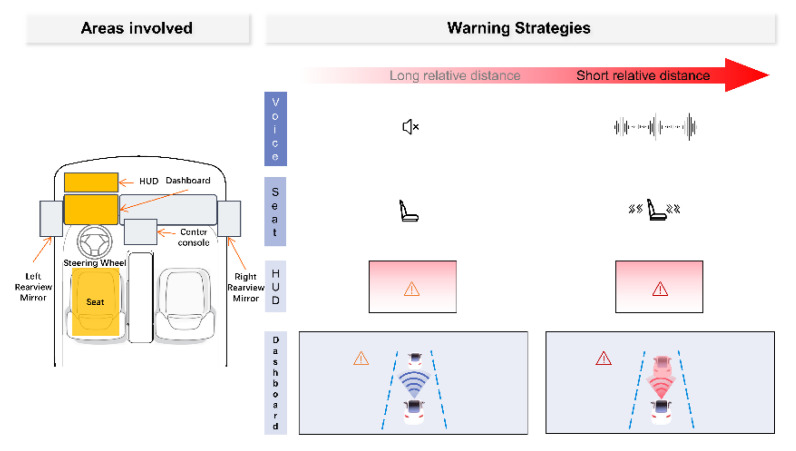
Design of multimodal cockpit of intelligent connected vehicle.

**Figure 2 sensors-22-01189-f002:**
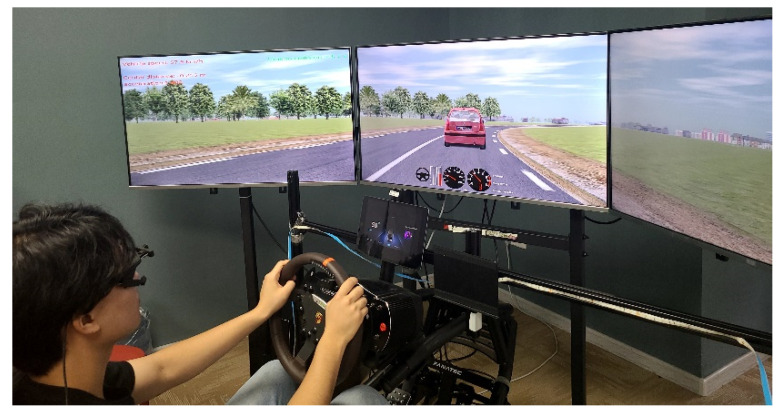
Forward collision warning experimental scenario and test process.

**Figure 3 sensors-22-01189-f003:**
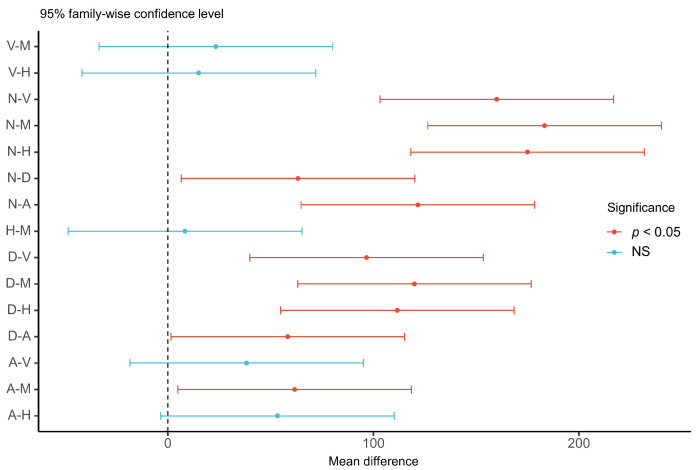
Post hoc test of reaction time in different warning modalities. V: seat vibration, M: multimodal, H: head-up display, N: no warning (baseline), D: dashboard, A: warning tone.

**Figure 4 sensors-22-01189-f004:**
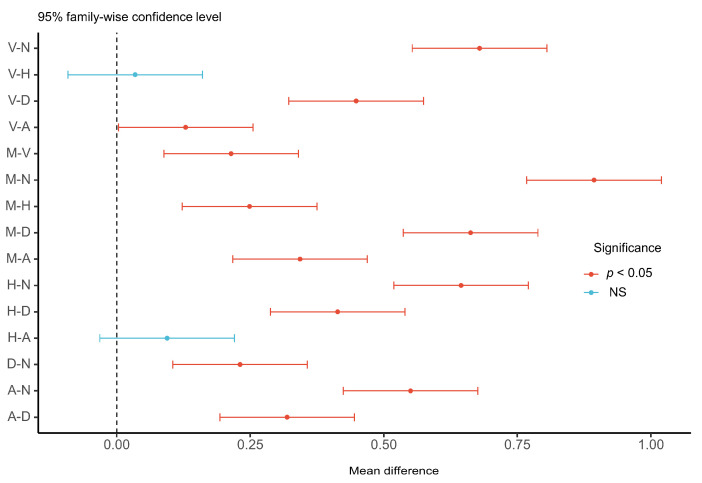
Post hoc test of time-to-collision in different warning modalities. V, seat vibration; M, multimodal; H, head-up display; N, no warning (baseline); D, dashboard; A: warning tone.

**Figure 5 sensors-22-01189-f005:**
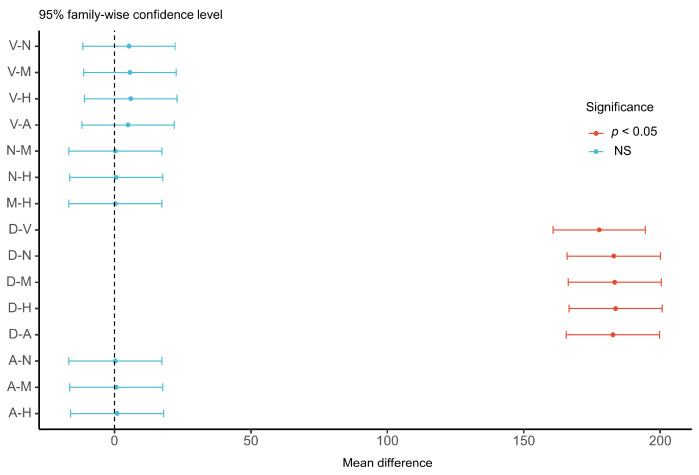
Post hoc test of off-road fixation duration in different warning modalities. V, seat vibration; M, multimodal; H, head-up display; N, no warning (baseline); D, dashboard; A: warning tone.

**Figure 6 sensors-22-01189-f006:**
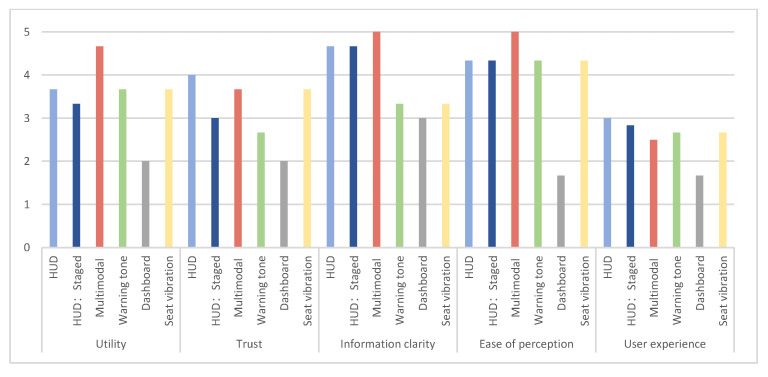
Subjective evaluation of warning strategies.

**Table 1 sensors-22-01189-t001:** Result of the ANOVA pre-test.

		Shapiro–Wilk Test		Levene’s Test
		W	*p*	*p*
Utility	Reaction time	0.98955	0.2099	0.1095
Driving performance	TTC	0.98958	0.2116	0.2056
	SDLD	0.98934	0.1970	0.4394
Visual workload	OFD	0.97928	0.2103	0.9660

**Table 2 sensors-22-01189-t002:** Summary of the utility in warning strategies.

Independent Measures		Utility
		Reaction Time
Modality	Baseline	631.67
	Dashboard	568.33
	Warning tone	510.00
	HUD	456.67
	Seat vibration	471.67
	Multimodal	448.33
Warning stage	Single-stage	733.33
	Multi-stage	490.00

The reaction time for the warning stage, timing starts at TTC = 5 s.

**Table 3 sensors-22-01189-t003:** Summary of the driving performance in warning strategies.

Independent Measures		Driving Performance	
		Time-to-Collision	The Standard Deviation of Lane Departure
Modality	Baseline	1.16	0.065
	Dashboard	1.39	0.066
	Warning tone	1.71	0.064
	HUD	1.80	0.065
	Seat vibration	1.84	0.065
	Multimodal	2.05	0.065
Warning stage	Single-stage	2.33	0.066
	Multi-stage	2.87	0.064

**Table 4 sensors-22-01189-t004:** Summary of the visual workload in warning strategies.

Independent Measures		Visual Workload
		Off-Road Fixation Duration
Modality	Baseline	164.67
	Dashboard	347.67
	Warning tone	165.00
	HUD	164.00
	Seat vibration	165.33
	Multimodal	164.33
Warning stage	Single-stage	164.00
	Multi-stage	165.00

The reaction time for the warning stage, timing starts at TTC = 5 s.

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
