# Peer review of "Evaluation of Multimodal and Multi-Staged Alerting Strategies for Forward Collision Warning Systems"

_sensors, 2022, doi:10.3390/s22031189_

Round 1
Reviewer 1 Report
Dear authors,
As following, some comments related to your paper:
- Overall, it seems you focused mainly on visual distraction; what about other types of distraction? For example, on page 3, section 2.1, when you write that “… if the information is presented on the center console or dashboard, the HMI will take up the driver's visual resources”.
- Can you provide more details about the driving simulator you used, in particular with the SCANER framework and related characteristics? Do you have (and, if it is so, have you used) the AD module?
- On page 5, lines 207-211, can you check the Unit of Measures? Are they correct?
- Same page, towards the end, in the randomization process, why didn’t you consider the baseline?
- On the end of page 8 and begin of page 9, you wrote that “… there was no significant difference in the standard deviation of lane departure …”, but this depends on the specific type of distraction you consider, in my opinion. Which is in your case? For example, visual-cognitive distraction can affect even the lateral behavior.
- On page 11, conclusion section, you wrote that “in the comparison between the multimodal warning and the baseline (no warning is given), there is no significant difference in vehicle lateral control …”, have you considered a scenario with distraction?
- Which are the next steps of your research activity you can plan (if any)? For example, the use of real-world data? Or, do you have already some application in mind?
I hope these comments can help to improve your paper.
Reviewer 2 Report
It is a good work, and it would be valuable to audience.
Reviewer 3 Report
This paper presents an interesting approach for multimodal and multi-staged alerting strategies in the case of V2X. This is indeed a
state-of-the-art track in automotive from the all the points that were presented in the paper as motivation.
In order to avoid forward collisions, an FCW system should provide timely and accurate alerts to the driver, and the alerts should not
significantly interfere with driving performance. This point was closely taken into consederation in the paper. The results are presented
in a professional matter and are supporting the goal of the research.
Some observations:
- The text for subchapter 3.3 should start on the same page as the name of the subchapter
